



# Enhanced weathering leads to substantial C accrual on crop macrocosms

Francois Rineau*[1], Alexander H. Frank[2,3], Jannis Groh[4,5,6], Kristof Grosjean[1], Arnaud Legout[7], Daniil I. Kolokolov[8], Michel Mench[9], Maria Moreno-Druet[1], Benoît Pollier[7], Virmantas Povilaitis[10], Johanna Pausch[11], Thomas Puetz[5], Tjalling Rooks[1], Peter Schröder[12], Wieslaw Szulc[13], Beata Rutkowska[13], Xander Swinnen[1], Sofie Thijs[1], Harry Vereecken[5], Janna V. Veselovskaya[8], Mwahija Zubery[1], Renaldas Žydelis[10], Evelin Loit[14]

[1]Environmental Biology, Centre for Environmental Sciences, Hasselt University, Diepenbeek, Belgium.
[2]Center of Stable Isotope Research in Ecology and Biogeochemistry (BayCenSI), BayCEER, University of Bayreuth, Universitätsstr. 30, 95445 Bayreuth, Germany
[3]Department of Plankton and Microbial Ecology, Leibniz Institute of Freshwater Ecology and Inland Fisheries (IGB), Zur alten Fischerhütte 2, 16775 Stechlin, Germany
[4]Department of Soil Science and Soil Ecology, Institute of Crop Science and Resource Conservation, University of Bonn, Bonn, Germany
[5]Agrosphere Institute (IBG-3), Forschungszentrum Jülich (FZJ), Jülich, Germany
[6]Isotope Biogeochemistry and Gas Fluxes, Research Area 1 "Landscape Functioning", Leibniz Centre for Agricultural Landscape Research (ZALF), Müncheberg, Germany
[7]INRAE, BEF, F-54000 Nancy, France
[8]Boreskov Institute of Catalysis, Novosibirsk, Ac. Lavrentiev av. 5, Novosibirsk 630090, Russia
[9]Univ. Bordeaux, INRAE, Biogeco, Bat B2, Allée G. St-Hilaire, F-33615 Pessac cedex, France
[10]Institute of Agriculture, Lithuanian Research Centre for Agriculture and Forestry, Lithuania
[11]Agroecology, BayCEER, University of Bayreuth, Universitätsstr. 30, 95445 Bayreuth, Germany
[12]Helmholtz Center Munich, Research Unit Environmental Simulation, Ingolstädter Landstrasse 1, D-85764 Neuherberg
[13]Institute of Agriculture, Warsaw University of Life Sciences, Nowoursynowska 166, 02-787 Warsaw, Poland
[14]Estonian Univ Life Sci, Inst Agr & Environm Sci, Tartu, Estonia

*Correspondence to*: François Rineau (francois.rineau@uhasselt.be)



**Abstract.** Enhanced weathering (EW) is proposed as a key strategy for climate change mitigation. It involves the application

of silicate rock powder to soils, where it is expected to react with $CO_2$ released from soil respiration, forming stable carbonate ions and thereby sequestering carbon. Here, we evaluated the effects of EW on a crop ecosystem within a macro-scale ecotron—an enclosed facility enabling complete quantification of carbon fluxes among the atmosphere, vegetation, soil, and leachates. EW treatment resulted in an almost three-fold enhancement of measured carbon flux into the soil, achieving rates up to 1.5 tons per hectare. Furthermore, the magnitude of carbon sequestration exceeded what could solely be attributed to

electrochemical transformations. Therefore, we conclude that EW facilitated significant carbon accrual in our simulated ecosystems via not only carbonate precipitation but also enhanced biogeochemical activities promoting additional carbon storage. Based on these findings, we speculate on the underlying pathways responsible for such outcomes.

## 1 Introduction

Agricultural systems are responsible for approximately 11% of anthropogenic $CO_2$ emissions (IPCC, 2014), highlighting the

urgent need for scalable mitigation strategies to meet the $CO_2$ reduction targets of the 2015 Paris Agreement. Enhanced rock weathering (EW) has emerged as a promising approach in this context (Beerling et al., 2020). The method involves applying silicate rock dust to soils, where it reacts with $CO_2$ from soil respiration to form bicarbonate and carbonate ions. Theoretically, these carbonates may either remain in the soil or be leached into groundwater and transported to aquatic systems, where they could help reduce ocean acidification—both pathways contributing to carbon sequestration. Early studies support some of

these theoretical predictions (Dietzen & Rosing, 2023; Guo et al., 2023; Rijnders et al., 2023), suggesting that EW could be implemented across large agricultural areas. Its carbon removal potential is estimated at 0.5 to 4 Gt $CO_2$ per year (Beerling et al., 2020), with additional benefits for soil health in nutrient-poor soils (Beerling et al., 2018). However, significant uncertainties remain—particularly for agricultural soils (Cipolla et al., 2021)—underscoring the need for robust experimental validation. Field experiments have revealed important findings, such as substantial $CO_2$ removal from soil pore water (Holzer

et al., 2023) and increased inorganic carbon accumulation (Kantola et al., 2023). However, the open nature of field settings complicates comprehensive tracking of ecosystem carbon fluxes, especially the fate of carbonate ions. Mesocosm studies, while more controlled, have often lacked sufficient soil depth (<70 cm), potentially missing important processes occurring deeper in the soil profile (Vienne et al., 2022). To address these limitations, we conducted an EW amendment experiment on an oat cropping system within an ecotron—a controlled, closed-environment facility that enables real-time monitoring of



ecosystem-level carbon fluxes (Roy et al., 2021). This setup allowed for a direct test of the hypothesis that basalt application

leads to carbonate formation and long-term carbon storage.

## 2 Material and Methods (as Heading 1)

### 2.1 Ecotron facility

At the UHasselt ecotron, large (4.7m$^3$, diameter 2 m) and deep (1.5 m) macrocosms are exposed to tightly controlled

environmental treatments in gas-tight enclosures, while high-frequency measurements allow accurate estimation of the

ecosystem C budget on an hourly scale (Rineau et al., 2019). This enables to study the actual dynamics of the process of EW

and the fate of its products in a real ecosystem, while the large macrocosms ensure a necessary degree of realism in the

measured processes (Clobert et al., 2018). In short, the macrocosms were placed within a large, gas-tight chamber topped with

a dome transparent to both UV and photosynthetically active radiation (PAR). Within the chamber, environmental conditions

including air temperature, precipitation, relative humidity, $CO_2$ concentration, and wind speed were precisely controlled. Soil

temperature and water tension were also regulated using a heat exchange system and a network of suction cups installed at the

base of each lysimeter—the vessel containing the soil and plant macrocosm. Each lysimeter was equipped with sensors located

at five soil depths (10, 20, 35, 60, and 140 cm) across three radial profiles (spaced 120° apart). These sensors continuously

monitored soil temperature, water tension, volumetric water content, and electrical conductivity. Above the plant canopy, air

samples were collected to monitor $CO_2$, $CH_4$, and $N_2O$ concentrations using two gas analysis systems (LGR 911-0011 and

SYNSPEC gas analyser; Envicontrol, Gembloux, Belgium). Additionally, net radiation (incoming minus outgoing radiation)

was measured using pyranometers, and PAR was monitored with a LI-190R quantum sensor. All measurements were recorded

at intervals ranging from once every 30 minutes to once per minute, depending on the parameter.

Soil water samples were collected *via* suction cups installed at 10, 20, 35, 60 and 140 cm depth and in triplicate in each

lysimeter. The tension in the suction cups was adjusted by a vacuum pump (VS Pro, METER group, USA) applying a constant

-50 hPa and -150 hPa to the upper (10, 20, 35 cm) and lower (60, 140 cm) cups, respectively, as upper soil layers are usually

drier. The water extracted from the cups was then stored in 1L-bottles sitting in a temperature-controlled cabinet (+10 °C).

Every three weeks, the water contained in the bottles was filtered at 0.45μm and analysed for total organic Carbon (TOC),

total Nitrogen (TN), anions and cations. Four water samples taken from the main pipe feeding the rain system were also filtered

and analysed in the same way, as well as water samples from the leachate tank (aliquot of the drainage).

## 2.2 Macrocosms

Six large macrocosms were extracted undisturbed from a dry heathland in 'Hoge Kempen' National Park, Belgium, in

November 2016 and placed in ecotron units at UHasselt. The soil in this plot is a brunic-dystric arenosol, with an organic layer

of 10-20 cm depth, on top of a sandy matrix containing 5-10cm clay lenses, and with a ferric iron precipitation horizon at, 150-

200 cm depth. The soil pH varies from 6 on the top, organic layer and 4 to 5 on the B horizon. The TOC content of the soil

varies between 1.9% in the top, organic layer and 0.5 % below down to 140 cm. They were exposed to recreated ambient

climate conditions until January 2020, after which they were "marginalized" by removing topsoil and vegetation to simulate

nutrient-poor marginal land. From January 2020 to September 2022, the macrocosms were subjected to a future climate

scenario (2070-2075) and agricultural treatments to mimic the conversion of heathland to crop fields. Barley was cultivated

with foliar Si amendments (Rineau et al., 2024), followed by mustard as a cover crop, and mineral NPK fertilization was

applied. Basalt was added to three units (10 t/ha) and incorporated into the topsoil, with all units tilled during seeding. The

basalt was finely ground (<1mm particle size) and consisted mostly of plagioclases and pyroxenes (38 and 26%, respectively;

for more details see Table S1). The basalt composition was moderately reactive for enhanced weathering, with relatively low

olivine (7%) and moderate feldspar (10%) contents, leading to a potential C removal of a maximum possible removal of 529

kg $CO_2$/t (144 kgC/t). We applied 10t/Ha, which theoretically led to a maximum of 1444 kgC/Ha of removal (Table S1). The

soil pH at that moment was 7 in all units. Oats were then planted and harvested after 150 days. Manual weeding was carried

out during the growing season. More details are provided in the "macrocosms details" box in the supplementary information.

## 2.3 Climate simulations

The experiment was conducted under climate conditions projected for 2070-2075 (RCP 8.5 scenario), allowing for realistic

simulations of elevated atmospheric $CO_2$ levels and their effects on the ecosystem. The ecotron served as a real-time, accurate

simulator of climate scenarios, controlling air temperature, humidity, rainfall, $CO_2$ concentration, wind speed, soil temperature,

and bottom soil water tension for each macrocosm. High-frequency monitoring was performed for key parameters, including

photosynthetically active radiation (PAR), net radiation, lysimeter weight, leachate weight, and concentrations of $CH_4$ and

$N_2O$. Soil water samples were collected at five separate depths and in triplicate for chemical analysis. More details about the

ecotron technicalities can be found in (Rineau et al., 2019; Roy et al., 2021). Climate data were downscaled to a half-hourly

resolution using projections based on local climate models (more details in (Vanderkelen et al., 2019)), and $CO_2$ levels were

adjusted to reflect future increases by adding a 221ppm offset to real-time measurements from a nearby ICOS station. The

characteristics of the climatic conditions applied to the six macrocosms are shown in Figure S1. The crops were exposed to a

climatic year typical of what is expected in 2070 according to local projections of the IPCC 8.5 scenario (Vanderkelen et al.,

2019).

### 2.4 Plant biomass

The quantification of plant biomass was conducted to assess the influence of basalt amendment on vegetation development as

well as to estimate the overall carbon balance within the ecosystem. At harvest, plant density was measured using three 50 x

50 cm quadrats per macrocosm. In each quadrat, oat stems were counted, and five plants (shoot and root) were randomly

sampled for biomass. Plants were dried at 60°C, and dry weights of grains, chaff, stems, leaves, and roots were measured,

along with the number of grains and leaves. Yield was calculated based on plant density and grain weight per unit area. Carbon

content in oat biomass was estimated by multiplying the dry weight of plant organs by stem density, assuming a 40% carbon

content (Sun et al., 2019). For weeds (primarily *Rumex acetosella*), biomass was measured after manual removal, corrected

for water content, and scaled to the lysimeter area, also assuming a 40% carbon content. Note that we tested the sensitivity of

our results to this C content assumptions, see the statistics section below.

### 2.5 $CO_2$-C net flux

Silicate rock amendments are of interest due to their potential to increase ecosystem carbon (C) sequestration. To verify this,

we estimated C sequestration in both plant biomass and soil by measuring net $CO_2$-C and $CH_4$-C fluxes. Net $CO_2$-C flux, the

largest component, was calculated as the exchange between the atmospheric compartment and each macrocosm. $CO_2$

concentrations in each chamber were maintained to mimic future levels (ambient + 221 ppm offset) by automated $CO_2$ gas

injection or scrubbing by extracting air from the chamber and passing it through a lime-filled container. The $CO_2$ levels were measured every 30 minutes by the means of an air sampler located 1m above the macrocosm's soil surface. The air sample is then conducted to a Synspec gas analyser where its $CO_2$ concentration is measured using gas chromatography. Given the

airtight nature of the chambers and the established relationship between $CO_2$ changes and injection or scrubbing time, we estimated the net $CO_2$-C stored or released over time. Fluxes were calculated every 30 minutes and converted to daily rates (g $CO_2$-C/m²) using the ideal gas law (air temperature on top of the macrocosm as well as air pressure are measured automatically). Data gaps were filled using a moving average function (ALMA function from the TTR package in R). Interventions in the chambers can slightly disturb the system's carbon balance, as $CO_2$ levels in the chamber equilibrate with

those of the main corridor. However, given the large chamber volume, we estimate that even a doubling of $CO_2$ concentration from 400 to 800ppm—assuming all additional carbon were taken up by the plants—would result in only about 50 mg of additional sequestered carbon per intervention. Over the entire experiment, this would amount to a maximum of approximately 1 g per unit, which is too little to affect the main conclusions of this study. Furthermore, the number of interventions was evenly distributed across units, ensuring that this effect was consistent between both treatments.

**2.6 CH₄-C net flux**

The net $CH_4$-C flux was determined as the exchange of $CH_4$-C between the atmospheric compartment and the macrocosm. $CH_4$ concentrations in the growth chambers were measured every 30 minutes using a Los Gatos Research (LGR) gas analyzer. Previous measurements confirmed that methane concentrations remained stable in the absence of a macrocosm and were unaffected by $CO_2$ scrubbing, but fluctuated rapidly when the chamber doors were opened. These events were therefore

identified and corrected for in the flux calculations. Additional tests verified that $CH_4$ fluxes were not influenced by microbial activity in the drainage system. The $CH_4$ budget was computed by correcting the $CH_4$ concentration changes for door-opening artefacts, then converting the resulting values from ppm to g $CH_4$-C $m^{-2}$ $day^{-1}$ using the ideal gas law, methane's carbon fraction (75%), and recorded chamber temperature and pressure.

**2.7 Rainwater C flux**

The amount of carbon (C) added to each macrocosm through simulated rainfall was estimated by multiplying the average non-purgeable organic carbon concentration in the irrigation water—based on triweekly measurements—with the cumulative precipitation over each corresponding three-week period. Precipitation amounts were estimated using the AWAT model (Hannes et al., 2015; Peters et al., 2017), following the approach described in (Rineau et al., 2024). These values were then aggregated for each macrocosm by summing the contributions over the entire growing season.

**2.8 Leachate C flux**

We estimated the amount of C leaching out of the macrocosm using the following calculation:

$$leachateC = leachate \times soil\ waterC$$

"Leachate" refers to the cumulative volume of water collected in each unit's weighable leachate tank during the three-week

intervals between sampling events, while "soil water C" denotes the average non-purgeable organic carbon concentration in the soil water samples collected at each sampling date. Based on prior experience, the values exceeding 50 mg/L were considered outliers and excluded from the calculations. Final values were aggregated by summing the carbon fluxes per unit over the entire growing season.

**2.9 Sampling C flux**

Some of the macrocosm C has also been exported through soil water sampling throughout the experiment. To account for this, we used the following calculation:

$$samplingC = 10 \times 12 \times average\ C$$

Where 10 corresponds to the average amount of water collected per unit per sampling campaign (10 l in every unit); 12 is the number of sampling campaigns during the experiment; and "average C" is the average non-purgeable concentration in this unit

throughout all samplings. The obtained values are therefore a C flux aggregated through the whole growing season.



**2.10 Soil C net flux**

In the previous sections, we described the measurements of the following fluxes: $CO_2$-C net flux, $CH_4$-C net flux, rainwater-C flux, leachate C flux, sampling C flux, and plant biomass C production. The only unknown flux needed to close up the C budget of the ecosystem is then soil C net flux, that was calculated as follows:

$$soilC = CO2C + CH4C + rainfallC - plantC - leachateC - samplingC$$

Where "soilC" is the soil C net flux; "CO2C" is the $CO_2$-C net flux (negative if the macrocosm was a net sink of $CO_2$-C); CH4C is the $CH_4$-C net flux (negative if the macrocosm was a net sink of $CH_4$-C); rainfallC the rainwater-C flux; plant-C is plant biomass C (positive), leachateC the leachate C flux and samplingC the sampling C flux. Since plant biomass was measured only once, at harvest, carbon fluxes could only be estimated as integrated values over the entire growing season.

Consequently, all related data were aggregated across the full experimental period (01/03/2022–01/09/2022). The net $CO_2$-C flux inherently accounts for both photosynthetic uptake and respiratory losses, and we were not able to differentiate between the two.

**2.11 Soil C pools**

To assess if EW resulted in the formation of inorganic C, we estimated the size of the different organic and inorganic C pools

(dissolved organic C: DOC, dissolved inorganic C: DIC, particulate organic C: POC and particulate inorganic C: POC). We monitored DIC by measuring the sum of carbonates, bicarbonates, and carbonic acid and DOC by measuring non-purgeable organic C in soil pore water collected from five depths (10, 20, 35, 60, and 140 cm) at three-week intervals throughout the experiment. For PIC and POC, we could do it only at the end of the experiment (because of constraints in sampling we could not harvest soil samples at the beginning of the experiment). So for these two measurements, if the treatment resulted in more

C in a given pool, we would measure a higher pool size in the basalt samples. At the end of the experiment, we collected 3 randomly taken soil samples per unit and dried them overnight in an oven at 60°C. Then, PIC was measured as total inorganic carbon (carbonates) in soil sample after drying, and was measured by measuring $CO_2$ formation after acidifying soil sample with phosphoric acid in TOC/TN analyser. Finally, POC was measured as total organic carbon on a TOC/TN analyser on dried soil samples.



### 2.12 C isotope analyses

We aimed to rule out the possibility that the basalt directly reacted with $CO_2$ from the atmospheric pool rather than with $CO_2$ derived from soil respiration. This process can be effectively traced using isotopic analysis, as the $CO_2$ supplied by the ecotron's control system has a distinctly negative $\delta^{13}C$ signature. If atmospheric $CO_2$ were directly fixed by the basalt, it would result in a detectable shift toward more negative $\delta^{13}C$ values in the soil carbon pool. With strong isotopically fractionating C-assimilation linked to photosynthesis, it is possible to trace and differentiate the plant-derived C, serving as a substrate to the soil organic carbon (SOC, $\delta^{13}C$ = - 27.0 ‰ vs. VPDB, see Figure S2) and being respired from $CO_2$-C, which might be directly adsorbed from the isotopically enriched signal being supplied into the ecotron via $CO_2$ tanks (-30 ‰ vs. VPDB). Thus, if the treatment with basalt would result in a shift of the isotopic ratio of the SOC towards the atmospheric value, this would be indicative of adsorption with further incorporation from this source into the SOC.

Soil samples were taken in September 2022, at the end of the experiment. They were dried (60°C, 24 h, atmospheric pressure) and subsequently milled using a ball mill (MM2, Retsch, Haan, Germany). Samples were then weighed into tin foil and measured against a lab standard (acetanilid, p.a., IVA Analysentechnik, Meerbusch, Germany) calibrated against international standards (IAEA CH-7 and IAEA CH-3, International Atomic Energy Agency, IAEA, Vienna, Austria) using an Elemental analyser (EA IsoLink CN, Thermo Fisher Scientific, Germany) connected via a continuous flow open split interface (ConFlo IV, Thermo Fisher Scientific, Bremen, Germany) to an isotope ratio mass spectrometer (DELTA V Advantage, Thermo Fisher Scientific, Bremen, Germany). All $^{13}C/^{12}C$ ratios are reported using the $\delta$-notation against the internationally recognized ratio of Vienna Pee-Dee Belemnite (VPDB) and expressed in per mille (‰) (Coplen, 2011; McKinney et al., 1950):

$$\delta^{13}C = (R_{sample}/R_{standard} - 1)$$

where R represents the ratio of the heavier to the most abundant stable isotope of the corresponding element (e.g., R = $^{13}C/^{12}C$), with $^{13}R_{VPDB}$ = 0.0111802, as recorded in the software (Isodat 3, version 3.0.94.12, Thermo Fisher Scientific, Bremen, Germany).

### 2.13 Effect of treatment on soil $CO_2$ adsorption capacity

Finally, we investigated whether the observed increase in soil carbon in the basalt-amended macrocosms could be attributed to a purely physical adsorption mechanism. Silicate rock powders possess a high surface area and a crystalline structure capable



of adsorbing various gases (Ramos et al., 2022), and could, in principle, retain $CO_2$ from soil respiration in the gaseous phase. To test this, we conducted a series of experiments on soil samples from the macrocosms to evaluate their $CO_2$ adsorption properties. We investigated whether the observed increase in soil carbon in the basalt-amended macrocosms could be attributed to a purely physical adsorption mechanism. We validated this assumption by conducting additional continuous-flow $CO_2$ adsorption experiments using a humid $CO_2$/Ar gas mixture (RH = 100%, $CO_2$ concentration = 28.5 vol%).

### 2.14 Microbial metabolic activity


Microbial metabolic activity in soil was measured to compare basalt and control treatments and to help interpret soil C fluxes. The FDA (Fluorescein Diacetate) assay, which estimates microbial activity by measuring the conversion of fluorescein diacetate to fluorescein by microbial enzymes, was used (Adam and Duncan 2001). For 13 consecutive days in May 2022, three soil samples (1 cm diameter, 10 cm deep) were collected per macrocosm, stored at -19°C, sieved, and aliquoted to 0.1

grams dry weight. Samples were incubated in sodium phosphate buffer with FDA solution at 30°C for 16 hours. After dilution, absorbance was measured at 490 nm, and values were converted to moles of substrate using a calibration curve.

### 2.15 Statistics

Data in Figure 1 summarize total carbon fluxes during the growing season, corresponding to single-time-point biomass measurements taken at harvest. However, hourly $CO_2$-C net flux recordings enabled finer-scale analysis, so we evaluated

treatment impacts on daily $CO_2$-C net flux using a linear mixed-effects model incorporating treatment, ecotron unit, and day as factors. Each flux is associated with its own standard deviation. Two components, "plantC" and "leachateC" were estimated through an internal bootstrap procedure to account for uncertainty arising from stem density estimates, the C content of plant organs (20% to 60%)(plantC) and the amount of water sampled per sampling event (5 to 15l). To propagate uncertainty across the entire soil C net flux calculation, we used a bootstrapping approach (n = 10,000) to generate empirical distributions of the

soil C net flux for each treatment. We then applied a bootstrap hypothesis test to evaluate the observed treatment effect against a null distribution assuming no treatment difference. Additional analyzes employed similar modeling approaches for other variables including organ-specific biomass, stem density, soil C isotopes, and inorganic C in soil water. For inorganic C in the soil solution (DIC), as well as Ca and Mg in the soil solution, as the dataset was gappy, but also more structured from the fixed

variables depth and date, we used a different bootstrapping approach. We generated empirical distributions of the concentration

of the element (iC, Ca or Mg) in soil solution at the unit level (in order to keep the structure induced by depth and date). We

then ran a generalized linear model with treatment, depth and their interaction as fixed variables and unit and date as random

variables, and computed the average estimate and p-value of the effect of treatment across all iterations (n = 1,000). Statistics

were done in R (R core team, 2019) using nlme (Pinheiro et al., 2023), dplyr (Wickham et al., 2023), lubridate (Grolemund &

Wickham, 2011) and vegan (Oksanen et al., 2013) packages.

## 3 Results

### 3.1 Climatic conditions

The spring began unusually cold with night frosts and temperatures below 5°C in early March, rapidly transitioning to warmth

in April ( 20°C maxima) and subsequent heatwaves in May ( 30°C). Summer remained consistently warm, punctuated by a

late-June heatwave. Rainfall was frequent yet light (<15mm/day) throughout most of the growing season except for brief

periods of drought and heavy rains in mid-May and early June. Consequently, soil moisture fluctuated between extremes: rapid

depletion to ~5% during the May heatwave, short-lived recoveries exceeding 10%, and sustained lows (<5%) throughout

summer (Figure S1).

### 3.2 Effect of basalt amendment on ecosystem C fluxes

The largest input flux, by far, came from net $CO_2$ exchange (366 to 457 gC/m$^2$ depending on the treatment), which was almost

two orders of magnitude larger than all other fluxes (1 to 7 gC/m$^2$ depending on the flux and the treatment) (Figure 1). The

crop system was a net $CH_4$ sink throughout the experiment. The macrocosm was a net C sink in both treatments  (456 gC/m$^2$

and 361 gC/m$^2$ in basalt and control units, respectively).



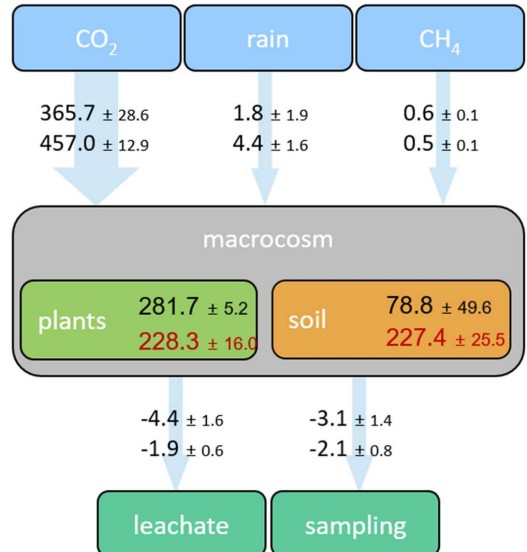

**Figure 1. Measured net C fluxes and pools to the macrocosms in both treatments (black: control, red: basalt), in g/m² aggregated for the whole growing season, and averaged per treatment (n=3; +/- standard error). The total macrocosm C pools entailed plant and soil C pools, plant C pool being measured directly and soil C pool calculated as a difference between total macrocosm C pool and the plant pool. Negative values indicate a net loss of C from the ecosystem, and positive values indicate net C gain in the ecosystem. Note that the sampling pool corresponds to the soil pore water sampling occurring every 3 weeks.**


We then partitioned the sequestered carbon (C) within each macrocosm into two primary pools: plant biomass and soil. Plant

biomass was significantly lower in the EW treatment (228 g C/m² vs. 282 g C/m² in the control) (Figure S3). Notably, the

estimated root biomass was relatively low, comprising only 6% of total plant dry mass (Figure S4), whereas values reported

in the literature typically range between 15% and 30% for oat (Gao et al., 2019). Despite thorough sampling, it is possible that

some root biomass remained in the soil, potentially leading to an underestimation. Based on this discrepancy, we estimated

that root biomass may have been underestimated by 10–59 g C/m².

Approximately 40% of the total fixed C was allocated to the soil pool (153 ± 41 g C/m² across treatments) (Figure 1). The

basalt treatment led to a substantial and statistically significant increase in soil C sequestration, with values 2.9 times higher

(227 ± 26 g C/m²) than in the control (79 ± 50 g C/m²), corresponding to a difference of 149 g C/m² (p = 0, bootstrap test; note

that the bootstrapped difference in mean values was 155 g C/m², see Figure S5). These results were robust to assumptions in





pore water sampling flux (5 to 15 L per sampling date; p = 0 and 149 to 150 g C/m² in soil sequestration) and plant C content

(20% to 60%; p = 0 and 133 to 197 g C/m² in soil C sequestration). Immediately after applying basalt, elevated $CO_2$ emissions

occurred likely due to soil disturbance (Figure 2). Subsequently, both treatments exhibited comparable $CO_2$ dynamics until

mid-May/mid-June when differences emerged, accelerating sharply during the June/July heatwave before stabilizing later in

the season (Figure 2). These trends coincided with rising soil temperatures (~20°C). Despite lacking separation of plant vs.

soil-derived carbon, negligible variations in final biomass suggest the observed patterns reflect differential soil carbon

sequestration. Notably, changes in net carbon balance were strongly inversely linked to soil temperature (Figure S6) rather

than moisture.

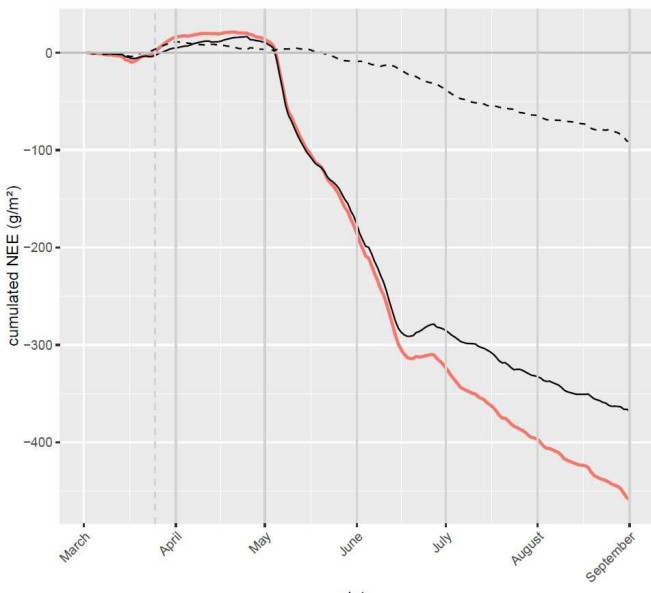

**Figure 2. Effect of the EW treatment on the dynamics of the C balance of the system. Black: control, Red: basalt treatment; dashed black line: difference between the two treatments; vertical dashed grey line: date of the basalt amendment. The NEE has been**
**estimated per hour and per unit (three units per treatment) using a model accounting for CO2 corrections, temperature and pressure, and aggregated per day (sum) and then per treatment (average) and cumulated throughout the growing season.  We also computed the difference between the average in the two treatments (dashed line).**



### 3.3 Soil carbonates in the basalt-treated macrocosms

No significant differences in inorganic carbon concentrations were observed between treatments at any depth, even when using

bootstrap approach to address the gappy nature of **the** dataset (estimate of treatment "control"= -0.72 +/- 1.49, average p-value=0.43) with both inorganic and organic C levels remaining below 0.2 gC/m² (1–5 mgC/L) (Table 1). To adjust for varying relevance of inorganic carbon increases across depths, we normalized values according to the respective soil column volumes and moisture contents. Specifically, an increment of 10 mg/L in shallow layers (short columns, lower moisture) contributed less significantly to the overall budget compared to deeper strata (longer columns, higher moisture). After multiplying

inorganic C values by adjusted soil volumes and moisture, bootstrapping yielded statistically nonsignificant results.(estimate of treatment "control"= 9.7 +/- 16.5 mgC/l, average p-value=0.43). Additionally, we re-ran the bootstrap analysis focusing exclusively on post–May 17th data, given the emergence of significant basalt-treatment effects on Net Ecosystem Exchange (NEE) thereafter. Nevertheless, this refined analysis still failed to reveal any discernible differences in dissolved inorganic carbon among treatments (estimate of treatment "control"= 21.2 +/- 23.3, average p-value=0.55). An alternate explanation

posits that carbonates might have initially formed in the soil but later degasseed as $CO_2$ during sample storage prior to analysis. Under this scenario, evidence of carbonate formation should manifest as elevated concentrations of divalent cations ($Ca^{2+}$, $Mg^{2+}$), resulting from silicate mineral dissolution. However, no significant treatment-related changes in calcium or magnesium concentrations were detected in soil pore waters (for Ca: estimate of treatment "control"= -3.17 +/- 3.28, average p-value=0.28; for Mg: estimate of treatment "control"= 0.14 +/- 1.76, average p-value=0.59, Figure S7).

One possible explanation for the absence of dissolved inorganic carbon is the precipitation of carbonate ions. Such carbonates would then be present as solid-phase PIC rather than in solution. To investigate this, we measured PIC in soil samples and found no significant differences between treatments (222 ± 13 g/m² in the basalt-amended plots vs. 214 ± 3 g/m² in controls; t-test, p = 0.28) (Table 2, Figure S8).




**Table 1. Inorganic carbon concentration of soil water samples taken by suction cups. The suction cups were installed at 5 different depths, and set at a tension of -150 HPa until 35cm deep and -50 HPa below. Soil water was sampled every 3 weeks and analysed for TOC and inorganic C.**

| unit | depth (cm) | 15/03 | 5/04 | 26/04 | 17/05 | 7/06 | 28/06 | 19/07 | 9/08 | 30/08 |
|---|---|---|---|---|---|---|---|---|---|---|
| 1 | 10 | | 15,6 | 8,8 | | 3,8 | 11,2 | 11,5 | | |
| | 20 | | 4,8 | 4,9 | 4,4 | 4,5 | | | | |
| | 35 | | 1,6 | 1,6 | 1,5 | 1,2 | 4,3 | 1,5 | | 4,5 |
| | 60 | | 1,3 | 4,5 | 0,7 | 0,6 | 4,3 | | | |
| | 140 | | | | | | 14,8 | | | |
| 3 | 10 | | 3,2 | 7,8 | | | | | 5,0 | |
| | 20 | | | | | | | 3,8 | 4,0 | |
| | 35 | | 1,5 | 1,5 | 0,7 | 4,5 | | | | 4,4 |
| | 60 | | 0,9 | 1,2 | 0,7 | 1,3 | 1,0 | | | |
| | 140 | 1,1 | | 0,9 | 0,8 | 0,8 | 0,9 | | | |
| 4 | 10 | | | 4,3 | 3,9 | 1,0 | 3,0 | | | |
| | 20 | | 4,5 | 4,3 | 3,5 | 1,7 | | 4,0 | | 4,0 |
| | 35 | | 4,5 | 3,4 | 1,4 | 5,3 | | | | |
| | 60 | | | | | | | | | |
| | 140 | | | | | | | | 7,2 | |
| Basalt | | 1,1 | 4,2 | 3,9 | 2,0 | 2,5 | 5,6 | 5,2 | 5,4 | 4,3 |
| 6 | 10 | | | | | 1,9 | | | | |
| | 20 | | | | | 6,2 | | | | |
| | 35 | | 4,7 | 0,8 | 3,6 | | | 3,6 | 4,9 | 4,3 |
| | 60 | 0,9 | 0,9 | 0,8 | 0,7 | 0,9 | 0,6 | | | |
| | 140 | 0,9 | 0,9 | 0,8 | 0,7 | 0,9 | 0,8 | | | |
| 10 | 10 | | | | | 3,9 | 5,5 | | | |
| | 20 | | 4,8 | 5,4 | 3,9 | | 4,1 | | 4,3 | 4,0 |
| | 35 | 1,2 | 1,5 | 1,6 | 1,2 | 5,0 | | 3,9 | 4,2 | |
| | 60 | 1,3 | 1,5 | 1,3 | 0,7 | 1,4 | | | | |
| | 140 | 1,9 | 1,9 | 0,9 | 1,5 | 3,3 | 5,0 | | | |
| 11 | 10 | | | 1,8 | 4,0 | 4,1 | 6,4 | | | |
| | 20 | | 5,1 | | | 4,3 | 4,8 | | | |
| | 35 | | | | | 0,8 | | | | |
| | 60 | | 1,0 | 0,8 | | | | | | |
| | 140 | | | | | | | | | |
| Control | | 1,2 | 2,4 | 1,8 | 2,3 | 3,5 | 3,1 | 3,8 | 4,5 | 4,2 |

**Table 2. Size of C pools in the topsoil (0 - 20 cm) of the macrocosms, in gC/m$^2$ measured at the end of the growing season. PIC: Particulate Inorganic Carbon, measured as soil dry mass in the topsoil (0-20 cm) samples. POC: Particulate Organic Carbon, measured as mmol/gDW in 20 cm deep soil samples. DIC: Dissolved Inorganic Carbon, measured as total C minus total organic C concentration (mg/l) in 10 and 20 cm soil water samples collected throughout the growing season by the lysimeter suction cups. DOC: Dissolved Organic Carbon, measured as organic C concentration (mg/l) in 10 and 20 cm soil water samples collected**
**throughout the growing season by the lysimeter suction cups. All four parameters were converted to gC m$^{-2}$ in the top 20 cm by accounting for a density of 1.3, 12g/mol C, and 10 % soil moisture for DIC and DOC (average moisture in the top 20 cm of soil across the growing season).**

| treatment | Basalt | | control | |
|---|---|---|---|---|
| | average | standard error | average | standard error |
| Soil PIC | 222 | 13 | 214 | 3 |
| Soil POC | 3713 | 720 | 5086 | 921 |
| Soil DIC | 0.03 | 0.03 | <0.01 | <0.01 |
| Soil DOC | 0.13 | 0.03 | 0.10 | 0.06 |

### 3.4 Contribution of soil C in the gas phase: $CO_2$ adsorbed to the basalt powder

Thermogravimetric analysis revealed that the presence of basalt approximately doubled the soil's $CO_2$ adsorption capacity.

However, desorption tests confirmed that the $CO_2$ was held through weak, reversible physical adsorption, with rates of 0.170 wt% and 0.036 wt% for basalt-amended and control soils, respectively (Table S2). These numbers were negligible in comparison with the other estimated C fluxes.

### 3.5 Direct fixation of atmospheric $CO_2$ by basalt

Finally, we aimed to rule out the possibility that the basalt directly reacted with $CO_2$ from the atmospheric pool rather than

with $CO_2$ derived from soil respiration. This process can be effectively traced using isotopic analysis, as the $CO_2$ supplied by the ecotron's control system has a distinctly negative $\delta^{13}C$ signature. If atmospheric $CO_2$ were directly fixed by the basalt, it would result in a detectable shift toward more negative $\delta^{13}C$ values in the soil carbon pool. However, we observed no such isotopic shift, suggesting that direct atmospheric $CO_2$ fixation by the basalt was negligible. Although isotopic variability was slightly higher in the basalt treatment, the overall pattern did not support significant atmospheric $CO_2$ incorporation (Figure

S2).

### 3.6 Microbial activity

Microbial activity was significantly higher (+32% on average, $p=0.005$) in basalt-treated soils midway through the growing season (Figure S9).

## 4 Discussion

### 4.1 ERW enhanced soil C pools substantially

Our study demonstrated a 2.9-fold increase in carbon sequestration under ERW treatment, adding 150 g/m² to the soil's carbon stock over one growing season, despite initial $CO_2$ emissions triggered by soil disturbance. Robust sensitivity analyzes supported this finding. Relative to baseline conditions, and based on the following reasonable assumptions (macrocosm volume of 4.4 m³, of a mean bulk density of 1.6 kg/dm³, in a 7 ton dry soil mass, and a mean total organic carbon (TOC) content of



0.7% (1.9% in the top 20 cm, 0.5% below, down to 140 cm depth)) this equated to a 1.2% boost in total soil carbon, far

surpassing the "4 per 1000" initiative's goal for mitigating climate change (Minasny et al., 2017).

This corresponds to an increase of 0.17 tons of carbon sequestered per tonne of basalt amendment applied (equivalent to 10

tons per hectare), falling within the range observed in other ERW field trials (Ramos et al., 2022; Swoboda et al., 2022).

However, most studies cited in these reviews involved amending with non-basaltic rocks and assessed performance over multi-

year timescales. A more analogous study by Kelland et al. (2020) similarly reported equivalent levels of carbon sequestration

using basalt, along with a comparable fold-increase versus controls (fourfold instead of threefold here). However, this outcome

was achieved under a considerably larger amendment dosage (100 tons per hectare). Alternatively, carbon sequestration rates

nearing 2 tons per hectare were realized within a single year using equal amendment quantities (10 tons per hectare), though

this involved utilizing a more chemically reactive substance—olivine—as opposed to basalt (Dietzen et al., 2019). Thus, the

quantity of carbon sequestered in our investigation notably exceeds typical expectations given the amendment type and dosage.

Let us explore potential influences on the observed carbon sequestration rates in our experiment. Firstly, the ambient $CO_2$

concentration was artificially raised by approximately 200 ppm as part of a broader research program investigating various

amendments' responses to projected climatic conditions (Rineau et al., 2024; Schroeder et al., 2021). Enhanced $CO_2$ levels are

known to stimulate both weathering rates and $CO_2$ uptake via electrochemically-driven reactions (Amann et al., 2022).

Additionally, elevated atmospheric $CO_2$ enhances crop biomass production (Kimball, 2016), amplifying carbon inputs into the

soil via root exudation. Secondly, we also measured significantly higher microbial activity in basalt-amended soils, mirroring

findings by Li et al. (2020), who reported a 33% increase in microbial activity following enriched rock-dust applications in

orchards. Such higher microbial activity can lead to localized elevations in $CO_2$ concentrations adjacent to amendment sites.

Thirdly, soil structures were preserved in our macrocosms, unlike the column-based study by Kelland et al. (2020), where

extensive soil sieving may have disrupted native soil structure, fauna and mycelium networks, thus emphasizing the importance

of soil organisms in sequestration processes. Fourthly, however, and most importantly, the degree of carbon sequestration

achieved—approximately 17% of the applied basalt amendment within a single growing season—is sevenfold higher than

comparable studies using basalt (Kelland et al., 2020). Remarkably, this figure slightly exceeds the theoretically achievable

limit of converting applied basalt into entirely carbonates (144 kg/ha for a 10 t/ha amendment). Hence, either complete

transformation of the amendment occurred exceptionally swiftly within three months, which appears improbable based on existing literature (Kelland et al., 2020), or alternative mechanisms contributing to soil carbon sequestration must be considered.

### 4.2 No evidence for inorganic C formation

This high C sequestration was moreover not followed by noticeable increases in soil inorganic C pools, contrary to expectations

with ERW. The experimental design did not allow us to take soil samples before the experiment and therefore to understand how different C pools were affected by the treatment, we can only compare the size of the pools in both treatments.

We did not detect any significant increase in soil inorganic C, whether in solution or as particulate inorganic C. We therefore have no evidence of carbonate ion formation, which is the central premise of ERW. The sequestered carbon was not detected in plant biomass or as gas trapped within amendment pores either. The most surprising is the lack of dissolved carbonates in

soil solutions. Due to gaps in soil-water chemistry data (particularly scarce samples from drier upper layers), a comprehensive carbon budget could not be constructed. Bootstrap analyzes indicated no intertreatment disparities in dissolved inorganic carbon, nor any evidence of ERW-relevant Ca/Mg ion leaching. Nevertheless, two plausible explanations remain for why bicarbonates might have gone undetected: first, potential sub-surface bicarbonate formation beyond our 20 cm measurement depth, although carbonate genesis typically initiates closer to the surface; second, extremely gradual topsoil carbonate

formation rates, particularly during the dry June–July period when most soil water was unavailable for sampling. While our current data provide no support for ERW-mediated reactions, we acknowledge methodological limitations prevent definitive conclusions regarding their occurrence.

### 4.3 Extra mechanisms of C sequestration

Nevertheless, the proposed bias fails to adequately explain the observed extent of soil carbon sequestration. Complete

dissolution of the amendment through ERW within such a brief timeframe is implausible and insufficient to account for the entirety of sequestration. Temporal dynamics of sequestration merit particular scrutiny: initiation coincided with topsoil temperatures crossing a 20°C threshold, occurring comparatively late in the growing season when significant plant biomass



had already developed. Intensification during a concurrent heatwave underscores thermal dependency. Limited soil moisture throughout the season reduced moisture-related constraints on EW reactions (Guo et al., 2023).

We speculate that under the amendment, root exudates interact with fine particles of the silicate rock, creating organo-mineral complexes that are making C recalcitrant. Indeed, weathering products of basalt are known to undergo such reaction, and these results could partially illustrate the synergy between EW and SOM accrual described by (Buss et al., 2024), despite the absence of clear evidence of EW in our study. Indeed it has been shown that mineral-associated organic matter (MAOM) is a strong stabilizing factor for organic matter (Cotrufo et al., 2012). However MAOM formation cannot account for all of the

sequestration observed, as it is dependent of formation of secondary minerals from ERW, especially since basalt is known to be more reactive for ERW than for SOM accriual (Buss et al., 2024).

Alternatively, basalt may stimulate root exudation, facilitating interactions with pre-existing soil constituents to produce particulate organic carbon (POM) or additional MAOM. However, root exudates constitute merely 5–10% of assimilated carbon (yielding a maximum contribution of 0.1–0.2 t/ha), which would contribute to only a fraction of the excess

sequestration. However, formation of aggregates due to abiotic process of Ca and Mg provided by the basalt also played a significant role in the sequestration potential, as was observed before by (Buss et al., 2024), especially since the basalt used in our study was 3 times richer in these elements than in that publication. However, according to the same study, plant should reduce this protection through plant exudates solubilising Ca and Mg (Buss et al., 2023). Root biomass was very low, suggesting that plant reduction of soil aggregation was limited. However, we did not observe clear solubilization of Ca and

Mg; but we cannot rule out this process for the same reason of low sampling intensity from end of May onwards. It is therefore possible that organic matter decomposition products may play a role as well there, and that labile organic C initially present in the soil and/or resulting from the decomposition substantially contributes to this very high sequestration rate.

### 5 Conclusion

Our results demonstrate that basalt addition enhances the crop ecosystem's role as a carbon sink, increasing soil carbon

sequestration by a factor of 3 over the growing season. It turned the macrocosms from low (0.5 tC/Ha)) to significant (1.5 tC/Ha) C sinks despite initial C loss due to soil disturbance when applying the treatment. We had no evidence that the ERW

reaction took place, but we cannot rule out that it happened because our experimental design did not allow for complete budget of dissolved inorganic C. However, the amount of sequestered C is too high to be explained only by ERW. We conclude that enhanced weathering led to substantial C accrual, possibly stimulating the formation of both mineral-asssociated and

particulate organic matter. Therefore, we advise caution in the widespread application of enhanced weathering until its effects on soil health and organic matter transformation are better understood.

**Author contribution**

FR, MM, VP, PS, WS, BS, RZ and EL contributed to conceptualization, methodology and validation. FR, AHF, JG, KG, AL, DK, MMD, BP, JP, TP, TR, XS, HV, JV, and MZ contributed to analysis and data curation. FR wrote the original draft. FR,

AHF, JG, AL, DK, MM, VP, PS, WS, BR, HV, JV, RZ and EL reviewed and edited the draft. FR, AHF, JG, AL, DK, TP, HV, JV contributed to the figures. FR, MM, VP, PS, WS, BR, RZ and EL contributed to the funding acquisition.

**Competing interests**

The authors declare that they have no conflict of interest.

**Acknowledgements**

We thank Carina Bauer and Petra Eckert of the Centre of Stable Isotope Research in Ecology and Biogeochemistry (BayCenSI, www.censi.uni-bayreuth.de) for their skilful technical assistance, and we thank Maria Sharro for assistance with graphical design of figures used in the manuscript. The authors used artificial intelligence tools (ChatGPT) to assist with English language editing of the manuscript.

**Financial support**

The study was supported by the FACCE-SURPLUS project BiofoodonMars, and financed by the Flemish Fonds voor Wetenschappelijk Onderzoek (FWO). D.I.K. and J.V.V. acknowledge the support by the Ministry of Science and Higher Education of the Russian Federation for Boreskov Institute of Catalysis (project no. FWUR-2024-0032 and FWUR-2024-0036)



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
