# Peer review of "Enhanced weathering leads to substantial C accrual on crop macrocosms"

_EGUsphere, 2025_

## Author Comment (AC8)

This paper reports results of a study of enhanced silicate weathering (EW) at an ecotron (enclosed macrocosm) facility, producing a dataset of C cycling in this system that is novel in this field (similar previous work has either been carried out in open-system field settings (e.g. Kantola et al. 2023), or mesocosms (e.g. Vienne et al., 2024) that are not enclosed). This allows for whole-system determination of fluxes under tightly controlled conditions that is crucial in order to address key outstanding questions in the field of EW – what is the short-to-medium-term fate of soil organic carbon when silicate fertilisers are added? What is the balance of inorganic carbon stored as SIC or DIC in pore water/leachate? And what is the overall GHG budget in these systems?

Comment#67: The study is valuable in this regard, but significant improvements must be made to the data presentation and interrogation here before this manuscript is fit for publication, to the extent that I would suggest a rejection and resubmission, rather than major revisions. Most glaringly, while the headline of soil C sequestration in the basalt-treated macrocosms is reported, this figure (calculated from difference in fluxes) is at odds with all of the data collected from the chemical pools in question (SOC, SIC, porewater DIC and DOC) — which in fact seem to show that the soil C pool is actually much greater in the control macrocosms rather than the basalt-treated macrocosms (driven by SOC, see Table 2). This discrepancy is not addressed at all, even though the authors do note in the discussion that the sink of C that their system-wide balance suggests cannot be traced by inorganic C formation. Figure 1 is therefore very misleading in this regard.

[...]

It seems to me that the big question here is how to reconcile the CO2 flux data with the other datasets the authors have generated. This must be addressed before this study should be resubmitted for publication – and my sense is that properly reconciling this discrepancy will result in a complete overhaul and re-write of the discussion, conclusions and headline takeaways.

We understand the reviewer to be suggesting a discrepancy between the reported carbon flux calculations and the chemical pool measurements, which appear to indicate greater carbon sequestration in the controls. However, we believe this is a misinterpretation of our results.

- -First, this table shows the size of the C pool at the end of the experiment: we assessed their absolute size rather than any temporal change. Since we could not measure these pools at the start of the experiment, it is possible that the control plots already had higher pool sizes initially.
- -Second, there is actually **no significant difference in POC pool size**. While the controls do show slightly higher values, the variability is large, as reflected in the standard errors. To the reviewer's credit, this point is not explicitly stated in the manuscript—it was included in an earlier version submitted to another journal but was removed following a previous reviewer's suggestion (in hindsight, we should have retained it).
- -Finally, and most importantly, as noted in the manuscript, given the relative size of these pools compared to the fluxes, along with the inherent variability of soil C content, **any increase in pool size** would have fallen within the range of measurement uncertainty.

For these three reasons there is no discrepancy between the pool size at the end of the experiment and the measured C fluxes.

We suggest to address this comment by adding this paragraph in the results section where the results of Table 2 are described:

"The results presented in Table 2 show the size of the C pools at the end of the experiment. Because pool sizes were not measured at the beginning of the experiment, it is possible that control plots already contained higher C pools initially. We did not observe statistically significant differences in POC pool size between the treatments. Although mean values appear higher in the controls, the variability is considerable, as indicated by the large standard errors. The relative size of these pools  $(3713 \pm 320 \text{ g/m}^2 \text{ in the basalt treatment and } 5086 \pm 921 \text{ g/m}^2 \text{ in the control; see Table 2), compared to the measured fluxes, together with the inherent variability of soil C content, means that increase in pool size due to the measured C sequestration fell within the range of measurement uncertainty."$

Comment#68: Note that there are several fluxes that are missing in their analyses. It seems to me that leachate was only analysed for DOC, not DIC. This may be significant (though previous studies, such as Kelland et al. 2020, which is cited by the authors, find that DIC export in leachate is negligible in mesocosm EW experiments).

It is true that we did not analyze the leachates for DIC, due to technical limitations related to the design of the leachate sampling system. Nevertheless, DIC would first need to be formed near the surface (as the basalt was applied to the top 20 cm) and should therefore have been detectable in the upper soil layers before reaching the lower lysimeter boundary. We did not find any evidence of this. But we agree that this deserves some clarification, that we think relates a lot to comment#1 (reviewer1). We suggest to address these two comments the same way.

However, and most importantly, the central conclusion of our study remains unchanged: the extent of C sequestration observed is significantly greater than what is theoretically possible through complete weathering of the added basalt, as calculated with the Renforth formula. In this context, whether or not some DIC was missed in the leachate is a secondary consideration.

Comment#69: The "Extra mechanisms of C sequestration" touched upon by the authors in the discussion are invoked because additional effects on top of weathering of mafic minerals charge-balancing cations with bicarbonate are needed to explain the difference in measured C fluxes. As such the authors focus on the recalcitrance of organic C here; but this ignores the fact that a change in proportion of OM as MAOM should not mean that SOC change cannot be measured in the soil. Furthermore, an explanation for greater C storage in soils as a result of EW is at odds with the observation (Figure S9) that microbial activity was higher in the basalt-treated macrocosms. This means more active respiration of SOC and should result in a greater CO2 flux out of the soil, as well as a reduction in SOC stock (which is in fact what the authors observe).

We agree with the reviewer. This was also pointed out by two other reviewers. We suggest to correct this as explained in replies to comments#2 and #12.

**Line-by-line comments:**

Comment#70: Line 88: Note that in the "marginalization" process, topsoil was removed. This makes the system deplete in TOC in the upper portion of the soil, which may be something to come back to in the discussion on measured C fluxes.

Indeed. We suggest to add these sentences in the discussion "It is important to note that these results were obtained on soil with initially low SOC, as the experimental site was established by converting heathland into cropland through removal of topsoil and vegetation. This initial condition likely influenced the processes observed. In particular, because the soil was far from C saturation, its capacity for additional C sequestration was relatively high. For example, mineral surfaces available for association with newly formed organic matter were more likely to be unoccupied, creating

conditions that favored stabilization of fresh C inputs. Thus, the experiment was conducted under circumstances representing a situation with one of the highest potentials for C sequestration." See also answer to comment#11.

Comment#71: Line 90: Future climate scenario (2070-2075) – what does this entail? (In terms of temperature, precipitation). [EDIT: I see this is given in Line 108;

Indeed, see line 108.

Comment#72: Line 91-92: How much foliar Si and NPK fertiliser was applied?

We appreciate the reviewer's comment. However, we believe this point has already been addressed in Rineau et al. (2024), where the amendment was applied to the plants rather than the soil, on the growing season preceding this experiement. For this reason, we consider it outside the scope of the present manuscript.

Comment#73: Line 92: To what depth was the basalt incorporated?

20cm. We will add this information in the manuscript. See also response to comment#30.

Comment#74: Line 93: Is there a particle size distribution for the basalt that you added? How was the basalt ground (e.g. ball mill?). Do you have data on the specific surface area of the basalt? These are all of consequence as the surface area distribution of the material is a key determinant of the reactivity.

Unfortunately we couldn't retrieve more information on the particle size distribution and the batch used in the experiment is now gone. See also reply to comment#31.

Comment#75: Line 95: Rephrase "...contents, and a maximum potential C removal capacity of 529 kg..."

We will correct accordingly.

Comment#76: Line 95: How was this C removal potential calculated?

Using Renforth formula. We will clarify in the new version of the manuscript. See also reply to comment#46.

Comment#77: Line 135: I'm not sure I follow this line of reasoning. As I read this now, this is trying to say the following: CO2 within the chambers was regularly measured and converted into fluxes. Following measurement (or continuously, I'm not sure I follow this), CO2 was either injected into the chambers or scrubbed from the atmosphere of the chambers in order to maintain the "future atmosphere" pCO2 of ambient + 221ppm. This process of amending the CO2 concentration in the chamber can interfere with C cycling within the chamber, as ambient CO2 concentration within the chamber does not replicate timescales of atmospheric mixing perfectly (?). Thus, to estimate the effect that this might have on calculated C fluxes, you assumed that the largest likely swing in CO2 concentration within the chamber as a result of amending CO2 is 400ppm, which – if completely taken up by plants and converted to biomass – yields 50mg of C, or 1g over the duration of experiment.

If this isn't quite accurate, I would sharpen the language used to make it clearer when exactly the CO2 amendment occurred relative to the measurement, and what you mean by the figure of 1g C during the experiment.

Thank you for this comment. We indeed realize that our original explanation may not have been sufficiently clear. What we intended to convey is that there may have been some confusion between three distinct aspects: (i) the CO2 control system, (ii) our model to estimate net ecosystem exchange, and (iii) potential disturbances caused by disruption of chamber seal integrity. Based on our observations, CO2 concentrations inside the chambers could increase from ~400 to ~800 ppm during a prolonged intervention. To provide a conservative estimate, we assumed that the entire excess of 400 ppm was not scrubbed by the Ecotron control system but instead fully assimilated through plant photosynthesis (which is unlikely in reality). Using chamber volume, average temperature, and pressure, this would correspond to approximately 50 mg of C fixed in plant biomass per intervention. Multiplying this by the number of interventions gives our estimate of about 1 g of potential error in C sequestration.

We suggest to replace 134-138 by the following text: "Interventions in the chambers can slightly disturb the system's carbon balance, as  $CO_2$  levels in the chamber equilibrate with those of the main corridor. During long interventions,  $CO_2$  concentrations inside the chambers were observed to increase from ~400 to ~800 ppm. For a conservative estimate of potential bias, we assumed that the entire excess of 400 ppm was not removed by the Ecotron control system but instead fully assimilated through photosynthesis, although this scenario is unlikely. Based on chamber volume, average temperature, and pressure, this would correspond to ~50 mg C fixed per intervention. Scaled to the total number of interventions, this results in an estimated maximum error of ~1 g C in sequestration".

Comment#78: Line 143: what is meant by "in the absence of a macrocosm"?

It literally means when no macrocosm was in the chamber. We wanted to verify that there was no other methane source outside of the macrocosm itself.

Comment#79: Line 145-146: What additional tests were performed to verify that microbial activity did not contribute to CH4 fluxes?

We temporarily sealed the drainage system, monitored CH4 levels, and compared them to presealing measurements. These details were not included in the manuscript, as they fall outside the main focus of this study, but we can provide this information if the reviewer considers it relevant.

Comment#80: Line 160: Does leachate C flux only relate to organic C? What about inorganic C (i.e. bicarbonate and carbonate alkalinity), which is likely to be a large flux of C?

See replies to comments#68 and #49. We did not detect any DIC in the 5 soil layers above the bottom of the lysimeter. We will clarify that the leachate indeed relates to organic C and that inorganic C is ruled out by higher depth measurements.

Comment#81: Line 187-188: What instrument/analyser did you use for DIC and DOC measurements?

We will clarify this in the manuscript.

Comment#82: Line 188-190: I would improve the wording of this for clarity. You measure concentration of Soil Organic Carbon and Soil Inorganic Carbon at the end of the experiment and then assess the difference between control and basalt-treated macrocosms to determine whether there has been an effect of the treatment on the formation and storage of SIC and SOC (I think this phrasing works a bit better).. I would not describe this as Particulate Inorganic or Organic Carbon, as this generally implies suspended load in solution (which is not something you have looked at, as far as I understand).

Indeed, the suggested wording is clearer. We will correct these sentences accordingly.

Comment#83; Line 193: Give the make and model of the analyser.

We will correct accordingly.

Comment#84: 2.12: "soil carbon" here refers to SOC+SIC, doesn't it, given the analytical procedure? I would make it clear in this section if these two C reservoirs were considered separately in the isotope mass-balance, or whether there were data suggesting it is possible to ignore the contribution of SIC.

Soil C indeed refers here to SOC and SIC. We will correct accordingly.

Comment#85: Line 244, 295: Replace "gappy" with "sparse", or change to "the dataset had gaps". I wouldn't use the word "gappy".

We will correct accordingly.

Comment#86: 3.1: Are climatic conditions allowed to vary within the chambers then? In Line 65, you say "environmental conditions including air temperature, precipitation, relative humidity, CO2 concentration, and wind speed were precisely controlled". This section implies that this is not the case. If so, I would explicitly state in Line 65 that environmental conditions (with the exception of CO2 concentration?) were allowed to vary with ambient conditions.

We may not fully understand this comment. Climate conditions in the experiment were controlled and followed a predefined climate scenario based on a climate model. In this section, we describe how the outcome of this scenario in the Ecotron units.

Comment#87: Figure 1: Should some of these values be in red font? E.g. CO2 net flux (there are two values in black).

Indeed. We will correct accordingly. See also response to comments#9 and #41.

Comment#88: Line 272-274: But important to state that this underestimate does not affect the EW treatment more than the control.

Thank you for pointing this out, we will add a sentence to clarify this.

Comment#89: Line 285: I understand that this is at a whole-system level; but this is not observed in the soil samples you measure (in fact the opposite). Surely it's better to amend this figure to show the actual measured values of soil C, and then note the discrepancy in C fluxes?

We understand that this comment refers to the values shown in Table 2. The POC values are not significantly higher in the control treatment (see the high standard error) and the sequestration rates are too low and integrated in a too short time to be reflected in the POC: they are lower than the noise. See also reply to comment#67.

Comment#90: Figure 2: Define NEE in the figure caption.

We will correct accordingly.

Comment#91: 3.3: Specify here whether the "inorganic carbon" that is being talked about is "dissolved inorganic carbon in soil pore water", or "soil inorganic carbon" – i.e. solid-phase carbonates. Note that the title "soil carbonates" implies the latter, when I think reading the paragraph suggests to me that the former is meant.

In this paragraph, we indeed meant dissolved inorganic carbon in soil pore water. We will correct accordingly.

Comment#92: Line 295: "the" is in bold where it shouldn't be.

We will correct accordingly.

Comment#93: Line 295: I don't understand what the figures quoted in brackets here are referring to – there seems to be a unit missing, and if this is a difference between treatments then it should be stated as such.

These are the difference in estimates of linear model after bootstrapping (hence expressed in mg/l). We will clarify this.

Comment#94: Line 304: Again, include units here.

We will correct accordingly.

Comment#95: Line 305: "Carbonates might have initially formed in the soil but later degasseed [spelling mistake] as CO2 during sample storage prior to analysis". How do the authors suggest this could have happened? Were samples heated or acidified prior to analysis? If dissolved, C would not have degassed but would be added to DIC.

We will correct the spelling mistake. We think that as the soil pore water samples underwent at least twice manipulations where they were exposed to low CO2 partial pressure in the headspace: when stored into the collection bottle for up to three weeks, when filtered and aliquoted to 20ml vials for storage. We cannot rule out that the carbonates dissolved in the water degassed at least partly as CO2 during these manipulations. We suggest to clarify by adding this sentence: "The soil pore water samples were subjected to at least two manipulations during which they were exposed to a low CO2 partial pressure in the headspace: first, during storage in the collection bottles for up to three weeks, and second, during filtration and aliquoting into 20 mL vials. Under these conditions, dissolved carbonates may have partially degassed as CO2."

Comment#96: Line 308-309: Again, include units here.

We will correct accordingly.

Comment#97: Figure S7: "effect was not significant at any depth" – to what confidence interval? Visually it does look like concentrations of Ca in solution were higher in basalt-treated samples as compared with control samples. I think what would be instructive here is to do a simple back-of-the-envelope calculation to look at effect size needed for mass-balance: assuming that all the difference in C flux between basalt and control is the result of DIC stored in soil porewater, and assuming that all of this DIC is in the form of bicarbonate charge-balanced by Ca and Mg ions, then how much of a change in the concentration of Ca in soil porewater would you expect to see? Is this change larger than the confidence interval?

Concentrations of Ca and Mg tended to be higher in the basalt treatment, but variability was substantial. A generalized linear model including treatment, depth, and their interaction as fixed effects, with unit and date as random effects, indicated that concentrations of both cations were not significantly affected by treatment at any level (p > 0.05), even after bootstrapping to account for the sparse dataset.

For a the back of the envelope calculation: assuming 150 g C m-2 of additional sequestered C in a 1.5 m deep soil column, with 10% volumetric water content in the top 60 cm and 30% in the bottom 90 cm, the total water volume would be approximately 330 L (acknowledging that water content varies throughout the growing season). At the same time, 380 L of leachate were collected over the season, corresponding to ~704 mg/L Ca equivalent. If we conservatively assume that C sequestration occurred at a constant rate over the growing season (rather than the observed peak in May–June), this would translate into an average difference of ~88 mg/L across 8 sampling dates. This value is about four times higher than the maximum concentrations measured in pore water and roughly 18 times higher than the seasonal average. Even accounting for the possibility that some of the reaction also releases Mg, the measured concentrations are still, on average, about one order of magnitude below the expected values.

Comment#98: Table 1: What does the colour-coding mean here? There are also no units for the concentration given here. I'm not sure that this data needs to be in the main text rather than the supplement; it is more instructive visually to show box-and-whisker plots (e.g. Figure S7).

As noted in our response to comment #43, we are happy to include this as a supplementary figure if desired. However, given that several reviewers raised questions regarding the pore water analyses, we believe it is important to present the full data structure and variability. A figure alone could give the misleading impression of a continuous, stable dataset. For information here are two possible figures:

Figure SX. Effect of treatment on inorganic C in soil solution (mg/l). Black: control, Red: basalt treatment. Grey area: 95% confidence interval. We performed a t-test to test the effect of basalt amendment on inorganic C in soil solution at every date, which returned no significant result.

Figure SX. Effect of basalt on inorganic carbon concentration of soil water samples taken by suction cups in function of depth. The suction cups were installed at 5 different depths, with 3 replicates per depth, and set at a tension of -150 HPa until 35cm deep and -50 HPa below. Soil water was sampled every 3 weeks and analysed for TOC and inorganic C. Values are pooled for all dates after basalt amendment until just before harvest (30/08/2022). The effect of basalt treatment has been tested by a mixed model with treatment as a fixed effect and unit as a random effect, and was not significant at any depth.

Comment#99: Table 2: I think the word "particulate" in this context is not necessary and is in fact misleading, as this does not refer to suspended OC in solution (which is the context in which it is usually used).

We will correct accordingly.

Comment#100: Line 330: 'wt%' cannot be described as a rate. Reword to clarify what you mean here.

Indeed, we meant amounts, not rates. We will clarify accordingly.

Comment#101: 3.5: I would carefully rephrase the writing here to be much more precise. Rather than "rule out the possibility that the basalt directly reacted with  $CO_2$  from the atmospheric pool rather than with  $CO_2$  derived from soil respiration", I would phrase this as "test to what extent atmospheric C was incorporated into soil inorganic C during the lifetime of the experiment" – note that this does not have to be mediated by basalt dissolution. Figure S2 does not allow you to observe a "shift", given that you do not test SIC before and after incubation in the atmosphere with negative d13C signature; though it does allow you to observe a difference (or lack of difference) between the

SIC C isotope signature between the two treatments, suggesting (as you say) that direct atmospheric CO2 fixation by basalt was negligible, as would be expected given the location of the basalt in the topsoil and the difference in CO2 concentrations in topsoils vs in atmosphere.

Thank you for the suggestion. We then suggest to replace this sentence by "Although Figure S2 does not allow us to observe temporal shifts in SIC  $\delta^{13}$ C (since we did not measure SIC before and after incubation under an isotopically distinct atmosphere), it does allow comparison between treatments. The lack of a meaningful difference in SIC  $\delta^{13}$ C between the basalt and control treatments suggests that direct incorporation of atmospheric  $CO_2$  into soil inorganic carbon via basalt was negligible. This result is consistent with expectations, given the position of basalt in the topsoil and the difference between soil and atmospheric  $CO_2$  concentrations".

Comment#102: Line 347: Yes, on a macrocosm-wide budget level – but crucially, are you able to measure the change in the actual soil C stock itself? Table 2 suggests the opposite trend! This is an important caveat that should be made here.

No, as the observed increase is within the measurement uncertainty of SOC measurements. See replies to comments#67 and #89.